statistics

replicability crisis, uncertainty, open science, interdisciplinary perspective, metaresearch

**Author for correspondence:**
Sabine Hoffmann
e-mail: shoffmann@ibe.med.uni-muenchen.de

# The multiplicity of analysis strategies jeopardizes replicability: lessons learned across disciplines

Sabine Hoffmann[1,2], Felix Schönbrodt[1,3], Ralf Elsas[1,4], Rory Wilson[6], Ulrich Strasser[7] and Anne-Laure Boulesteix[1,2,5]

[1]LMU Open Science Center, [2]Institute for Medical Information Processing, Biometry, and Epidemiology, Medical School, [3]Department of Psychology, Psychological Methods and Assessment, [4]Institute for Finance and Banking, Munich School of Management, and [5]Department of Statistics, Faculty of Mathematics, Computer Science and Statistics, Ludwig-Maximilians-Universität München, Munich, Germany
[6]Research Unit Molecular Epidemiology, Institute of Epidemiology, Helmholtz Zentrum München German Research Center for Environmental Health, Neuherberg, Germany
[7]Department of Geography, University of Innsbruck, Innsbruck, Austria

SH, 0000-0001-6197-8801; FS, 0000-0002-8282-3910; RE, 0000-0001-8117-1175; US, 0000-0003-4776-2822; ALB, 0000-0002-2729-0947

For a given research question, there are usually a large variety of possible analysis strategies acceptable according to the scientific standards of the field, and there are concerns that this multiplicity of analysis strategies plays an important role in the non-replicability of research findings. Here, we define a general framework on common sources of uncertainty arising in computational analyses that lead to this multiplicity, and apply this framework within an overview of approaches proposed across disciplines to address the issue. Armed with this framework, and a set of recommendations derived therefrom, researchers will be able to recognize strategies applicable to their field and use them to generate findings more likely to be replicated in future studies, ultimately improving the credibility of the scientific process.

## 1. Introduction

In recent years, the scientific community has been rocked by the recognition that research findings often do not replicate on independent data, leading to what has been referred to as a replication crisis [1], reproducibility crisis [2] or statistical crisis in science [3].

In particular, a series of attempts to reproduce the results of published research findings in different disciplines found that these replication efforts produced much weaker evidence than the original study [4–7]. It has been estimated that in preclinical research alone, approximately $28 billion dollars are spent every year on research findings that are not replicable [8]. The crisis has consequences far beyond an insular world of scientists. Experts with strongly disagreeing viewpoints and publicized results that are subsequently contradicted are highly detrimental to the trust the general public and decision makers have in scientific results. This distrust endangers one of the key functions of science—providing robust research findings that can be built upon to help tackle important challenges to society [9]. The recent intense public debate surrounding key 'global issues' identified and targeted by the United Nations [10]—such as, for example, climate change and migration—and the ambivalent public perception of scientific contributions to these issues illustrates the importance of the scientific community retaining credibility.

While there have been a number of widely publicized examples of fraud and scientific misconduct [11,12], many researchers agree that this is not the major problem in this crisis [3,13]. Instead, the problems seem to be more subtle and partly due to the multiplicity of possible analysis strategies [14,15].

For a given research question of interest, there is usually great flexibility in the choice of analysis strategy, as many possible strategies are acceptable according to the scientific standards of the field [16,17]. The resulting multiplicity of possible analysis strategies is nicely illustrated by two recent experiments performed by Silberzahn *et al.* [18] and Botvinik-Nezer *et al.* [19]: Silberzahn *et al.* [18] recruited 29 teams of researchers with strong statistical background and asked them to answer the same research question (Are football referees more likely to give red cards to players with dark skin than to players with light skin?) with the same dataset. Similarly, Botvinik-Nezer *et al.* [19] invited 70 independent teams to test nine hypotheses on a single neuroimaging dataset. In both experiments, the teams obtained highly varied results, as they approached the data with a wide array of analytical techniques. There is evidence that the combination of this multiplicity with selective reporting can systematically lead to an increase in false-positive results, inflated effect sizes and overoptimistic measures of predictive performance [15,20–23]. Ignoring the multiplicity of analysis strategies can therefore lead to an overconfidence in the precision of results and to research findings that do not replicate on independent data.

While the social and biomedical sciences have been at the heart of the recent replication crisis in science, the multiplicity of analysis strategies has also contributed to credibility crises in other disciplines, e.g.—very prominently—in climatology. In 2009, e-mails and documents of leading climate scientists at the University of East Anglia became publicly available. Taken out of context, parts of these e-mails suggested that researchers felt it was 'a travesty' they could not 'account for the lack of warming', and included an allusion to 'Mike's Nature trick' to 'hide the decline' [24]. This incident, which received broad media attention, became known as 'climategate' and led to an erosion of belief in climate change [25,26] by creating the impression that climatologists are exploiting the multiplicity of possible analysis strategies to obtain overly alarmist results.

In response to the current crisis in science, a myriad of solutions to improve the replicability of empirical findings have been developed in different disciplines. There are for instance a number of recently proposed approaches which assess the robustness of research findings to alternative analytical pathways by reporting the results of a large number of analysis strategies: the 'vibration of effects' approach in epidemiology [27], 'specification curve analysis' [28] and 'multiverse analysis' in psychology [29], a 'measure of robustness to misspecification' in economics [30] or 'multimodel analysis' [31] and 'computational robustness analysis' [32] in sociology. In other disciplines, including climatology, ecology and risk analysis, there is a long-standing tradition of addressing the robustness to alternative analysis strategies through sensitivity analyses, multimodel ensembles [33] and Bayesian model averaging [34,35]. While the development of approaches addressing the issue of the multiplicity of possible analysis strategies remains important, we currently risk 'reinventing the wheel' in each discipline. In order to avoid the proliferation of approaches that address the same problems with similar ideas, we consider it advisable to benefit from lessons learned in other disciplines by means of a multidisciplinary perspective. In this work, we define a framework on common sources of uncertainty arising in computational analyses across a broad range of disciplines, covering both the statistical analysis of empirical data and the prediction of complex systems of interest through mechanistic physically-based models. The aim of this framework is to provide a common language to efficiently translate ideas and approaches across disciplines. We illustrate how it can help researchers benefit from experiences gained in other fields by giving an overview of solutions and ideas that have been proposed to improve the replicability and credibility of research findings.

# 2. The multiplicity of analysis strategies: examples from epidemiology and hydroclimatology

In a large number of disciplines, an important part of a given research project is the generation of numerical results describing the association between $p$ input variables (denoted by $X_1, X_2, \ldots, X_p$ in the following) and an outcome variable $Y$ through a mathematical function $f()$. $f()$ is typically referred to as the model, while the input variables $X_1, X_2, \ldots, X_p$ are called independent, exogenous or explanatory variables, or predictors, features, attributes or covariates, depending on context and discipline. The outcome variable $Y$ is also known as the dependent, endogeneous or response variable, or the output, label, criterion or predictand. In the following, we refer to this type of research, which relies to some extent on data, as empirical research, in contrast to research that is of a purely theoretical nature.

To illustrate the multitude of analyst decisions necessary in empirical research, we consider two examples from different disciplines. The first example is the analysis of data from an epidemiological study on the link between meat consumption and the risk of colorectal cancer to answer a research question in public health which has attracted considerable attention in recent years. The second example, from hydroclimatology, concerns the prediction of water mass stored in seasonal snowpack and its release as meltwater into the river runoff [36,37], which is essential in the prediction of future flood occurrence and water availability for irrigation and hydropower generation. Figure 1 illustrates how the analysis decisions in empirical research, applied to these two examples, lead to a multiplicity of analysis strategies.

Both in observational epidemiology and hydroclimatology, the first step is to collect data on the phenomenon of interest. In our example from epidemiology, these data come from $n$ individuals who are assumed to form a representative sample from a specified population of interest. In our example from hydroclimatology, the system to be investigated is a valley of a certain size in which the winter snowpack is transformed into spring snowmelt-induced streamflow. The data, which are indexed by space and time, consist of two parts. The first part is historical data for which observations at one or more meteorological stations exist for the input variables which include e.g. time-series data of measured air temperature and precipitation covering the previous 20 years. The outcome is the gauge streamflow at the outlet of the valley. The second part consists of values of the model input variables which reflect future changes in temperature and precipitation resulting from different greenhouse gas emission scenarios: with these model input variables in hand, and a developed model, one can predict the future evolution of the seasonal snowpack and hence, the resulting streamflow.

For both examples, before beginning analysis, the data must be preprocessed, a procedure involving numerous subjective choices. The flexibility in data preprocessing partly arises because the research hypotheses are generally not precise enough that they fully specify the input and the outcome variables [15,38,39]. Indeed, while measuring meat consumption and determining incidence of colon cancer may naively appear to be straightforward, the analyst has considerable flexibility in the definition of these two variables. For the same research question, one could consider meat consumption of all kinds, focus on red meat or processed meat, or distinguish between beef, pork, lamb and chicken [40]. Similarly, concerning the outcome, it is possible to concentrate on colon cancer, on rectal cancer or to include all types of colorectal cancer and even precancerous lesions like colorectal adenoma. In our example from hydroclimatology, we are faced with similar choices. For instance, in the absence of measurements of the input variables for all locations in the region of interest, the recordings from a single meteorological station have to be extrapolated. Furthermore, the possible values of temperature and precipitation reflecting changes in greenhouse gas emission scenarios are themselves outputs of mechanistic models. There are a large number of strategies to obtain both spatial extrapolations and these possible future values.

Following data preprocessing, we next decide on a model to describe our phenomenon of interest. In epidemiology, the aim is to control for all variables which might confound the association between meat consumption and colorectal cancer. These variables may for instance include body mass index, smoking, physical activity, socio-economic status and the consumption of alcohol, fruit and vegetables [41,42]. However, this is not an exhaustive list and there is no clear guidance on which variables should be considered mandatory in the model and which will lead to an unnecessarily complex description of the phenomenon of interest. Similarly, a model describing how snow accumulates, is stored and melts can be based on a variety of alternative model assumptions, potentially leading to '1701 snow models' [36]: examples include modelling the snow microstructure and evolution over time in physical detail, or a more simplified description of these processes in a snowpack representation with a single layer. Additionally, there are a number of model parameters to be specified in order to predict the seasonal

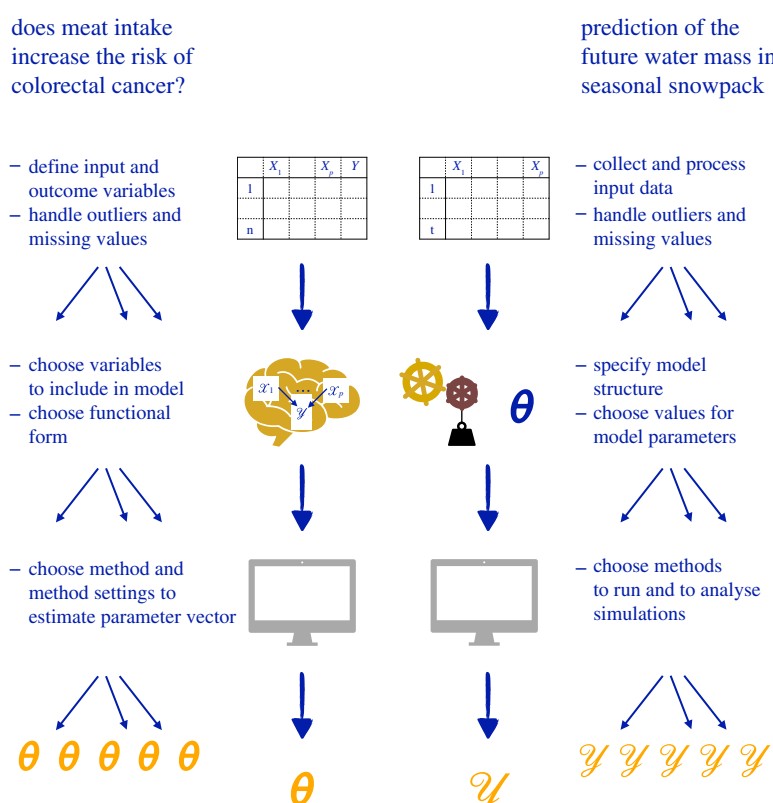

**Figure 1.** The multiplicity of analysis strategies arising from data preprocessing, model and method choices to obtain an estimate of the parameter of interest $\theta$ and values of the outcome variable $\mathcal{Y}$ for two research questions in epidemiology and hydroclimatology, respectively.

snowpack evolution, parameters such as the surface albedo and roughness length, and the initial density of the snow [37].

Once the model structure and its constituent variables are specified, there remain further decisions concerning the method to use to obtain the main result of interest. In our example from epidemiology, the result of interest is an estimate of a parameter $\theta$ describing the increase in colorectal cancer risk associated with meat consumption. Common estimation techniques for this model parameter include maximum-likelihood estimation, Bayesian inference, least-squares minimization and the method of moments. In our hydroclimatology example, the result of interest is the outcome $\mathcal{Y}$, i.e. the water equivalent in the seasonal snowpack and its melt. A first method choice concerns the discretization of the system to be investigated in space and time, i.e. the temporal and spatial resolution of the model set-up. Further method choices arise due to the terabytes of simulation outputs typically produced. These simulation outputs consist of a myriad of state variables describing for instance snow depth, snow surface temperature and snow density. To be able to interpret these outputs, they have to be aggregated, analysed and illustrated; however, different transformations and spatial and temporal aggregation techniques may either mask or accentuate oscillations and trends which may be present in these results.

Through the illustration of two examples from different fields, we see that for a research question of interest, researchers across disciplines are faced with a multitude of choices when presented with data in a situation where there is no clear guidance on analysis strategy from a theoretical or a substantive point of view. Although some of the choices made could be considered 'wrong', many would also be justifiable. As all justifiable paths are likely to lead to different results, we see there is a source of variability attributable to the choice of analysis strategy. In the presentation of scientific results, however, this variability is not commonly accounted for or discussed.

# 3. Sources of uncertainty arising in empirical research

As the aim of research is to expand existing knowledge by operating on the edge of what is known, it is hardly surprising that there are numerous sources of uncertainty arising in scientific discovery. In this

section, we will introduce a general, albeit inevitably incomplete, framework on common sources of uncertainty arising in computational analyses and show how the combination of these sources of uncertainty with selective reporting can lead to unreplicable research findings.

The idea behind the use of a mathematical model $f()$ is, in general, either explanation or prediction [43,44]. The modelling of the association between meat consumption and colorectal cancer in epidemiology can be seen as an instance of explanatory modelling. The main aims of explanatory modelling are to test a causal hypothesis [20], i.e. to assess to what extent a theoretical variable $\mathcal{Y}$ is influenced by the theoretical variables $\mathcal{X}_1, \ldots, \mathcal{X}_p$. As these theoretical variables are not directly observable, they have to be operationalized by defining measurable outcome $Y$ and input variables $X_1, X_2, \ldots, X_p$ [44]. Once these observable variables are measured on a sample of observations, statistical methods can be used to estimate the value of an unknown parameter of interest $\theta$ which quantifies the association between $Y$ and $X_1, \ldots, X_p$. The reporting of this parameter estimate is typically combined with a $p$-value and a confidence interval, which are used to test the research hypothesis concerning the association between $\mathcal{Y}$ and $\mathcal{X}_1, \ldots, \mathcal{X}_p$.

Our example from hydroclimatology, the modelling of the future evolution of the seasonal snowpack, on the other hand, can be seen as an instance of what can be referred to as mechanistic predictive modelling. The idea of mechanistic predictive modelling is to predict the values of an outcome $\mathcal{Y}$ at new or untried values of the input variables. In contrast to explanatory modelling, the goal of mechanistic predictive modelling is hence to apply a model to predict the behaviour of a system which is so complex that it would be difficult to predict and analyse otherwise [45]. Mechanistic predictive models (also referred to as physically- or process-based models [46,47]) typically heavily rely on subject matter knowledge and the (physical) principles underlying the behaviour of the studied phenomenon. Hence, model predictions are derived by relying on a number of physical laws or mechanistic assumptions, and the values of a certain number of model parameters $\theta = (\theta_1, \ldots, \theta_p)$, which are assumed to be known.

With the increasing availability of large datasets and improvements in computational efficiency, a second type of data-driven predictive modelling coming from an algorithmic modelling culture—often from artificial intelligence and more specifically from machine learning—is growing in popularity in many disciplines [48,49]. Here, the function $f()$ is estimated by an algorithm rather than by fitting a pre-specified model class to the data [43]. These algorithms try to dispense with (potentially restrictive) assumptions on the association between $X_1, X_2, \ldots, X_p$ and $Y$ [50] and do not typically rely on theoretical reasoning: they can thus be referred to as 'agnostic' predictive models.

Despite the main focus of explanatory modelling being the estimation of an unknown parameter and the main focus of predictive modelling being prediction, many analyses are concerned with both aims. For example, some parameter values in mechanistic predictive modelling can be determined by fitting the model to historical data in which both the input variables and the outcome are measured, a process referred to as calibration [45]. Conversely, in explanatory modelling, where the main focus is on explanation, the estimated parameter values can be used to predict new observations and to evaluate the adequacy of the chosen probability model [20].

While certain types of models are more popular in some disciplines than in others, there is no unique assignment of disciplines to modelling strategies. Mechanistic predictive models, which are popular in disciplines ranging from the geosciences and risk analysis to decision analytic modelling in health economics, can also be used in the prediction of infectious disease dynamics in epidemiology [51]. Explanatory modelling, on the other hand, which is popular in disciplines such as biology, psychology and economics [44,52], can for example also be applied in climatology when assessing the extent to which an extreme event can be attributed to anthropogenic climate change [53].

As illustrated in our examples from epidemiology and hydroclimatology, the multiplicity of possible analysis strategies can arise from data preprocessing, parameter, model and method choices. *Data preprocessing uncertainty* is caused by all decisions needing to be made in the selection of the data to analyse and in the definition, the cleaning and the transformation of the input and the outcome variables. Additionally, there is usually *model uncertainty*, as the best or the 'true' model structure to describe the phenomenon of interest is unknown. *Parameter uncertainty*, which is mainly present in predictive modelling, arises through model parameter values having to be specified when the analyst is armed with neither precise theoretical knowledge nor direct measurements [45]. In mechanistic predictive modelling, estimates of these parameter values can be achieved by observation of the system, through calibration or incomplete expert knowledge [45,46], but substantial uncertainty regarding the true values typically remains. Similarly, the performance of many algorithms for agnostic modelling is sensitive to hyperparameters [17], which have to be specified before running these algorithms. Examples

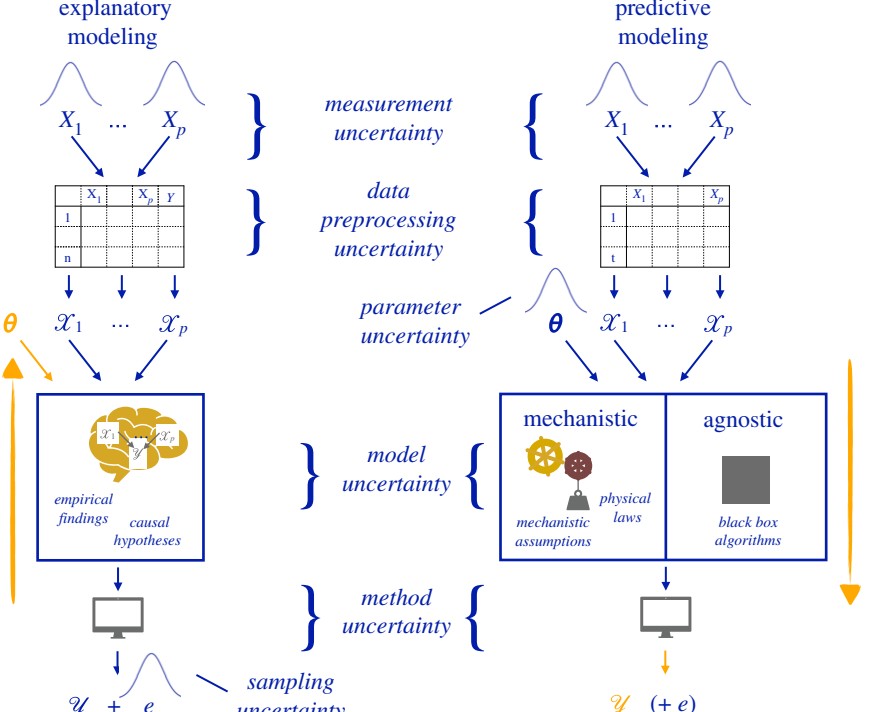

**Figure 2.** Sources of uncertainty in explanatory, mechanistic predictive and agnostic predictive modelling. Data preprocessing, parameter, model and method uncertainty are epistemic sources of uncertainty arising from a lack of knowledge in the specification of the analysis strategy. Measurement and sampling uncertainty are random sources of uncertainty that lead to variability in the results when the same analysis strategy is applied on different datasets. The model structure describes the association between the $p$ input variables $X_1, X_2, \ldots, X_p$ and the outcome of interest $Y$. $\theta$ is a parameter and $e$ represents a probabilistic error term.

include the minimal size of splitting nodes in random forests, the kernel and the cost parameter in support vector machines, and the number of neighbours in k-nearest neighbours [54]. As there is typically no clear guidance on which values to choose for these hyperparameters, their specification is sometimes considered to be more of an art than a science. Finally, researchers encounter *method uncertainty*, as specifying a model and parameter values is not sufficient to run the actual computations. Before the parameters in a statistical model can be estimated or predictions from a mechanistic or agnostic model derived, researchers need to choose, or even to develop, a specific implementation and computational method. Again, there is a multitude of options without clear guidance or a definitive choice on the method that will provide the most suitable answer to their research question [55].

The uncertainties detailed above—data preprocessing uncertainty, parameter uncertainty, model uncertainty and method uncertainty—are epistemic: they arise due to a lack of knowledge. As illustrated in figure 2, these epistemic sources of uncertainty can be contrasted with two additional sources of uncertainty, namely *measurement uncertainty* and *sampling uncertainty*.

*Measurement uncertainty* is ubiquitous in empirical research as it is generally impossible to determine the input variables $\mathcal{X}_1, \ldots, \mathcal{X}_p$ and the outcome $Y$ with absolute precision and accuracy. Depending on the discipline, information on these variables may be acquired through questionnaires, measurement devices or experimental protocols, which are all, to some extent, prone to imprecision. Finally, *sampling uncertainty*, which is especially prominent in explanatory modelling, results from the variability introduced when analysing a dataset assumed to be a random sample from a larger population of interest. This variability is often expressed through an error term $e$. Table 1 provides a short description of all six sources of uncertainty.

The interplay between these random sources of uncertainty and the multiplicity of analysis strategies arising from the four epistemic sources of uncertainty can lead to unreplicable research findings when combined with selective reporting, as illustrated in figure 3. If there is no restriction on the chosen analysis strategy, a researcher may try to compare the results of many strategies—each a path resulting from the given preprocessing, model, parameter and method choices—and then select the final analysis strategy based on the 'nicest' result: a smaller $p$-value, an effect in the 'desired' direction,

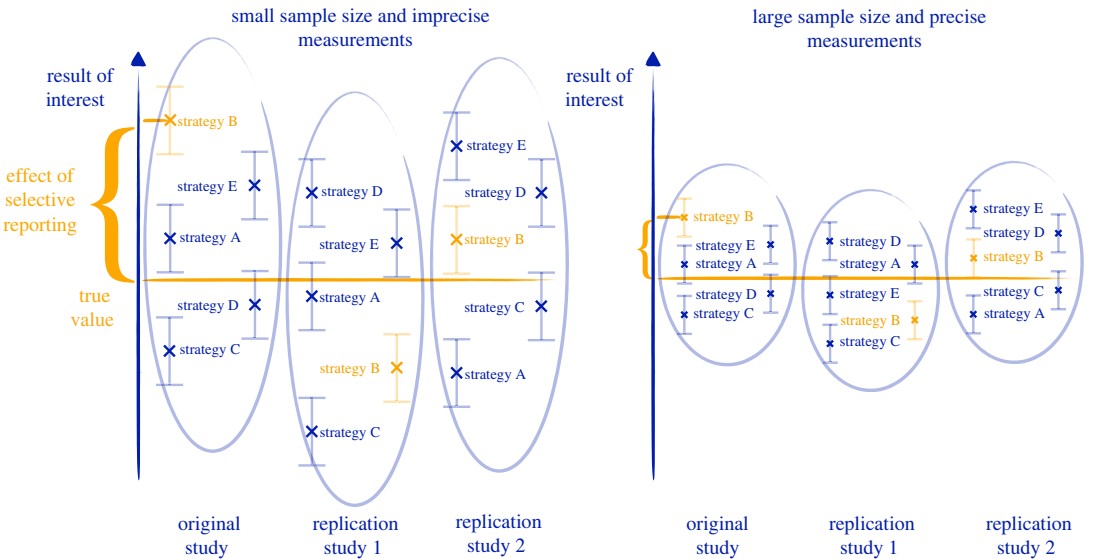

**Figure 3.** The impact of random sources of uncertainty and of the multiplicity of possible analysis strategies on the replicability of research findings. The result of interest is the parameter $\theta$ in explanatory modelling, the outcome $\mathcal{Y}$ in mechanistic predictive modelling and the predictive performance in agnostic predictive modelling. The yellow colour represents the results of the chosen analysis strategy—a strategy selected because it presents the most 'favourable' results. It is clear that the traditional confidence interval (given by the bars around the estimate 'x'), which only takes into account sampling uncertainty, is inadequate in capturing the true uncertainty in the estimate.

**Table 1.** Description of the six sources of uncertainty arising in empirical research.

|  | description |
| --- | --- |
| measurement uncertainty | randomness arising from the operationalization or the measurement of the input and the output variables |
| data preprocessing uncertainty | uncertain decisions in the selection of the data to analyse and in the definition, the cleaning and the transformation of the input and the output variables |
| parameter uncertainty | uncertain decisions in the specification of input parameters |
| model uncertainty | uncertain decisions in the specification of the model structure to describe the phenomenon of interest |
| method uncertainty | uncertain decisions in the choice of a method and method settings |
| sampling uncertainty | randomness arising from the selection of a sample from a larger population of interest |

or predictions in accordance with the expectations of the researcher, for example. This 'selective reporting' can lead to a substantial overestimation of the result of interest in empirical research, an effect heightened by small samples and imprecise measurements [1,56].

In explanatory modelling, confidence intervals and hypothesis tests typically only account for sampling uncertainty, while ignoring measurement and epistemic uncertainty. This limited focus thereby leads to apparently precise results which are not robust to variations in the choice of analysis strategy; they therefore have a high probability of being contradicted in a replication study.

# 4. Lessons learned across disciplines

The solution to the replication crisis which has probably received the most attention in the scientific community and beyond is the abandonment of statistical significance within the scientific literature and its replacement with Bayes factors, confidence intervals or other inferential methods [57–61]. While there may be inferential paradigms that are easier to interpret and less prone to overconfidence

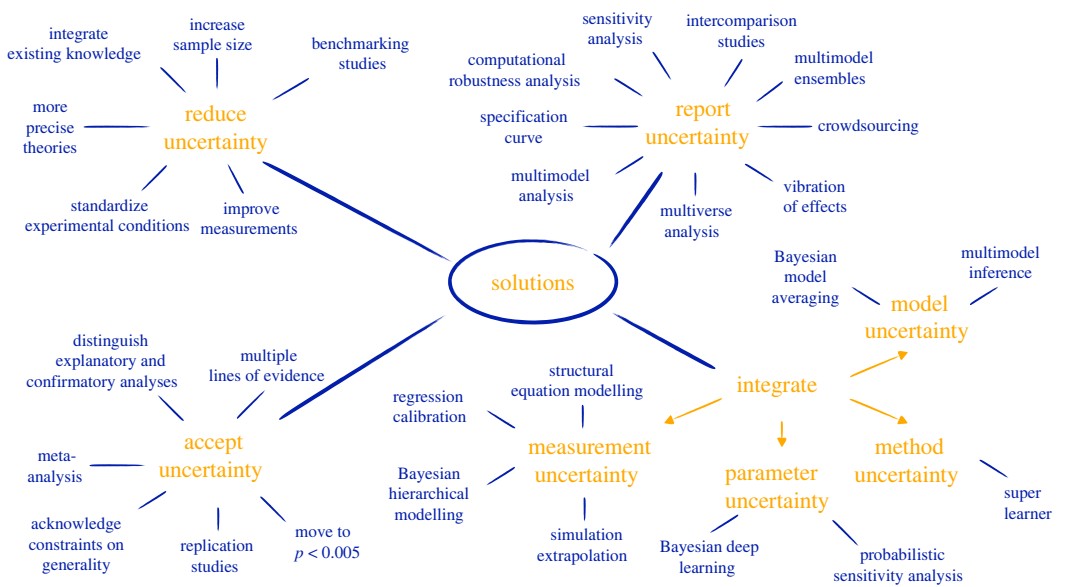

**Figure 4.** Overview of solutions to the replication crisis which address the multiplicity of analysis strategies by reducing, reporting, integrating or accepting uncertainty. For an interactive version of this graphic with assorted references see https://shiny.psy.lmu.de/multiplicity/index.html.

than null hypothesis statistical testing [62,63], the simple solution of jettisoning statistical significance can seem somewhat shortsighted in addressing the issues. Taking confidence intervals as an example, figure 3 illustrates how they can be just as prone as $p$-values to selective reporting. More generally, no inferential paradigm in itself is immune to overconfidence and the result-dependent selection of an analysis strategy from among a multiplicity of possible strategies [64,65].

Another prominent solution which has been proposed in response to the replication crisis in science is to increase transparency by promoting reproducible research practices [66,67]. While the publication of research data, code and materials can help build trust in science [68] and make the entire research process more efficient, it only indirectly addresses the multiplicity of possible analysis strategies. Transparency alone is again not enough to prevent selective reporting or eliminate overconfidence in results [69].

We therefore argue that we have to go beyond open science practices and the focus on statistical significance as the main culprit in the non-replicability of research findings by explicitly addressing the sources of uncertainty introduced in the previous section. A wealth of ideas and approaches to reduce, report, integrate or accept one or several of the sources of uncertainty have been discussed in the literature, leading to a myriad of solutions in different disciplines. In figure 4, we give an overview of these solutions.

## 4.1. Reduce uncertainty

There are a variety of strategies to reduce one or several sources of uncertainty. In explanatory modelling, a reduction in sampling and measurement uncertainty can for instance be achieved by increasing the sample size of studies [70–72], by improving the quality of measurements [1] or by standardizing experimental conditions [73–75]. To reduce model and data preprocessing uncertainty, Steegen *et al.* [29] and Schaller [76] call for more conceptual rigour and precise theories to reduce the number of possible analysis strategies. Method uncertainty, on the other hand, can be reduced through adoption of the results of 'benchmarking' studies, which aim to identify a best method for a given research question of interest in a given setting [55,77]. The integration of existing knowledge into explanatory modelling can also reduce uncertainty and help to obtain more precise parameter estimates. An example is the specification of informative prior distributions [78] in Bayesian inference, where the prior evidence can range from functional information in genome-wide association studies [79] to historical data in clinical trials [80].

## 4.2. Report uncertainty

In many disciplines, there is a long-standing tradition of reporting the results of a large number of possible analysis strategies, or the variability of these results, to assess their robustness to alternative assumptions

and model specifications. Common examples of this strategy include extreme bounds analysis in econometrics [81], multimodel ensembles [33] in hydrology and climatology and sensitivity analyses, which are used across many disciplines. More recent approaches to report uncertainty include the 'vibration of effects' approach [27], 'specification curve analysis' [28], 'multiverse analysis' [29] 'multimodel analysis' [31] and 'computational robustness analysis' [32], as discussed previously. Silberzahn *et al.* [18] go a step further and propose the reporting of the results of different teams of researchers analysing the same research question on the same dataset. In their 'crowdsourcing approach', it is thereby possible to simultaneously report the variability in results due to data preprocessing, model and method uncertainty, as different teams of researchers are likely to follow different paths in formulating their analysis strategies. Considered from a multidisciplinary perspective, this approach is similar to intercomparison studies, which have a long tradition in mechanistic predictive modelling in hydroclimatology [82]. In cases where the data are made publicly available, methods to report uncertainty can be applied by readers and reviewers to assess to what extent the originally reported results are robust to alternative analysis choices.

## 4.3. Integrate uncertainty

There are a number of approaches which can generate broader and more realistic uncertainty intervals by integrating measurement, model, parameter or method uncertainty when deriving parameter estimates in explanatory modelling and predictions in mechanistic and agnostic predictive modelling. In explanatory modelling, it is possible to account for measurement uncertainty through, for example, structural equation models [83], Bayesian hierarchical approaches [84], simulation extrapolation or regression calibration [85]. With regard to model uncertainty, Bayesian model averaging [34,35] and multimodel inference [86] go beyond the simple reporting of the results of all possible models by weighting the parameter estimates or predictions of all candidate models to produce a single summary measure and a measure of its uncertainty. In mechanistic predictive modelling, Bayesian melding [87] and probabilistic sensitivity analysis [88] can be used to integrate parameter uncertainty. Similarly, in agnostic modelling, it has been suggested to account for parameter uncertainty through Bayesian deep learning, where the uncertainty in hyperparameters is described by a prior distribution [89], and to integrate method uncertainty by combining the weighted predictions of a great number of candidate methods through a so-called 'Super learner' [90].

## 4.4. Accept uncertainty

Many authors have argued that classical statistical methods used in explanatory modelling suggest a disproportionate level of certainty [59,91] and that the replication crisis in science is in fact a 'crisis of overconfidence in statistical results' [92]. In this sense, a solution to the current crisis is to acknowledge the inherent uncertainty in scientific findings. This can be achieved by recognizing that statistical inference within exploratory analyses should be interpreted with great caution and that scientific generalizations need to be based on cumulative knowledge rather than on a single study [92]. Strictly confirmatory analyses can be realized either through the pre-registration of analysis plans [93,94] and registered reports [95], where the analysis strategy is specified in detail before observing the data; or through blind analyses, where researchers select an analysis strategy while being blinded to the outcome of interest [96]. Alternatively, it is common in agnostic modelling to perform exploratory and confirmatory analysis on the same dataset through split analysis plans: one part of the data is used to determine the best analysis strategy, the other to fit the final algorithm and determine its predictive performance [52].

A focus on cumulative evidence can be found in calls for replications as post-publication quality control [15,97,98] and in the proposal of Benjamin *et al.* [99] to redefine statistical significance by considering a $p$-value of $<0.05$ merely suggestive (i.e. having to be confirmed in subsequent studies) and only $p$-values $<0.005$ significant. In psychology, Simons *et al.* [100] emphasize the need for cumulative evidence and encourage authors to specify a 'constraints on generality' statement, which clearly identifies and justifies the target population of reported research findings. Cumulative knowledge can also be achieved by providing multiple lines of convergent evidence, also referred to as triangulation [101,102]. In biology, these lines of evidence can stem from several independent experiments, experiments performed for instance with isolated molecules, in cultured cell lines, or using animal models. Lastly, in psychology and medicine, we see the usefulness of meta-analyses, the summarization and aggregation of the results of similar studies.

**Table 2.** Six steps researchers can take to make their research findings more replicable and credible.

|  | steps |
|---|---|
| before the analysis | (1) be aware of the multiplicity of possible analysis strategies |
|  | (2) if possible, reduce sources of uncertainty in the study design |
| during the analysis | (3) if possible, integrate remaining sources of uncertainty into the analysis |
|  | (4) report the results of alternative analysis strategies |
| after the analysis | (5) acknowledge the inherent uncertainty in your findings |
|  | (6) publish all research code, data and material |

# 5. Steps to take to make one's own research more replicable

Based on the lessons learned across disciplines discussed in the previous section, what are the steps an individual researcher can take to improve the replicability and credibility or his or her own research? In table 2, we derive six simple steps researchers can take to make their own research more replicable and credible. A first step, which should not be underestimated, is simply to be aware of the multiplicity of possible analysis strategies and the potential for selective reporting. As pointed out by Nuzzo [103], even the most honest researcher is a master of self-deception and it is easy to jump to conclusions when finding patterns in randomness.

Once aware of the potential for increased uncertainty, one should evaluate and implement possibilities to reduce both the randomness in the data and the flexibility in analysis plan. In our example from epidemiology, we could for instance have reduced some of the sources of uncertainty by determining an adequate sample size through a power calculation, by integrating results from previous studies on meat consumption and colorectal cancer to specify an informative prior distribution and by clearly defining the research hypothesis and the input and outcome variables before collecting the data. In our hydroclimatology study, measurement and model uncertainty could have been reduced by, for example, including a very large number of confirmed measurements of the input variables, e.g. through the integration of remote sensing data, and by using only models that had been extensively validated elsewhere.

When deriving the result of interest, one should attempt to integrate all sources of uncertainty which could not be reduced in the previous step. Since to this point, there is no all-encompassing method accounting for model, measurement, method, data preprocessing and parameter uncertainty simultaneously, an alternative is to systematically report the robustness to alternative analysis strategies through one of the approaches presented in the last section. The next step—again, not to be underestimated—is to acknowledge the inherent uncertainty in the presented research findings and thereby avoid misleading readers into overinterpretation of the relevance of the results.

Finally, to make one's research findings more credible and improve the efficiency of the research process as a whole, one should publish all research code, data and material, both to allow others to try alternative analysis strategies and for reuse of the data in future studies.

# 6. Conclusion

Despite growing evidence for its pervasive impact on the validity of research findings, current research practices largely fail to address the multiplicity of analysis strategies. Currently, it is a highly profitable strategy to analyse small datasets and to exploit the multiplicity of possible analysis strategies arising from data preprocessing, model, parameter, and method uncertainty to obtain significant and surprising results. These results have a high probability of getting published, but a low probability of being replicated in subsequent studies. In the short term this lack of replication may simply be embarrassing, but in the long run this strategy has devastating consequences for the scientific community. While imprecise but convergent results are often readily accepted by the public, multiple apparently precise but contradictory results have a negative impact on the credibility of research findings [104,105]: these contradictory results can easily be discredited as conflicting evidence to create the impression that scientific knowledge is unreliable and that there is no scientific consensus on important research topics [106]. According to van der Linden *et al.* [107], this line of argumentation

has been used for years to delay or prevent regulatory actions concerning climate change, maybe contributing to the low belief in anthropogenic climate change among the American public [108].

If, on the other hand, we address the multiplicity of possible analysis strategies arising through data preprocessing, model, parameter and method uncertainty through reporting, integrating and acknowledgement, we will obtain broad but more realistic measures of our uncertainty, and research findings that are robust to the choice of the analysis strategy.

It is important to raise awareness of the fact that the multiplicity of possible analysis strategies is an issue affecting many different disciplines in similar ways; this awareness will enable us to join forces in our efforts to increase the transparency, replicability and credibility of research findings. Integrating multidisciplinary experience and insights is not only essential in the further development of appropriate solutions and in the elaboration of guidelines to help researchers make their research more replicable, but also in generating enough momentum to bring about change. As long as the reward structure in academia favours significant, overly clear-cut, and hypothesis-consistent results, researchers might be tempted to exploit the multiplicity of possible analysis strategies instead of addressing this issue in a transparent way to make research findings more replicable. This creates a social dilemma structure where societal and scientific interests are at odds with the individual career interest of researchers.

The multiplicity of possible analysis strategies is likely to become an even bigger challenge with the advent of increasing amounts of data that are not originally recorded for research purposes in many disciplines, for instance, in the form of routine care data in medicine, of administrative data in the social sciences and of remote sensing data in ecology [20,109–111]. These data are not the result of well-designed experiments where we have accurate knowledge on the data generation process, and a small set of research hypotheses of interest. Instead, the data may be imperfect, heterogeneous, noisy and high-dimensional [49,112]. When analysing these data, sampling uncertainty, which has attracted a disproportionate amount of attention in the scientific community to this point, will be comparably small, but measurement, data preprocessing, model and method uncertainty will be much larger than when dealing with more traditional data [113].

Given the importance and the urgency of the challenges we are facing today, we need scientific results that are veracious—both in their precision and in their (preliminary) imprecision. Novel and exciting but unreplicable results impede scientific progress and its societal translation. By addressing the multiplicity of possible analysis strategies through the framework and approaches suggested here, we can make the research process more efficient and improve the replicability, and ultimately the credibility, of research findings.

Data accessibility. This article has no additional data.

Authors' contributions. S.H. drafted the manuscript. A.-L.B. initiated and coordinated the project. F.S., R.E., R.W. and U.S. substantially contributed to the conception of the project and revised it critically for important intellectual content. All authors gave approval for publication and agree to be accountable for all aspects of the work in ensuring that questions related to the accuracy or integrity of any part of the work are appropriately investigated and resolved.

Competing interests. We declare we have no competing interests.

Funding. The project was partly funded by individual grant BO3139/7-1 from the German Research Foundation (DFG) to A.-L.B.

Acknowledgements. We thank Alethea Charlton for helping us to prepare the manuscript and Ulrich Mansmann, Jack Bates, Ansgar Opitz, Ute Hoffmann and Ronan Le Gleut for their comments.

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
