## [Peer Review File · Royal Society Open Science]

Review History

RSOS-201925.R0 (Original submission)

Review form: Reviewer 1 (Amy Orben)

Is the manuscript scientifically sound in its present form?

Yes

Are the interpretations and conclusions justified by the results?

Yes

Is the language acceptable?

Yes

Do you have any ethical concerns with this paper?

No

Have you any concerns about statistical analyses in this paper?

No

Recommendation?

Accept with minor revision (please list in comments)

Comments to the Author(s)

The manuscript integrates understanding across multiple disciplines to present a framework around analysis multiplicity. It does so with, what I think is, complete success. I thought this manuscript was carefully written, clearly argued and of importance to a broad scientific audience. I would therefore recommend it for publication.

There are a couple of minor points that the authors can decide whether they think would improve the manuscript:

1. I would suggest the authors might want to lengthen their figure headings slightly (e.g. Fig. 1 and 2), as readers often read those separately to the paper and at the moment they are not self-explanatory.
2. On page 7 there were two instances where the structure of sentences made the argument less clear than it could have been. For both "method uncertainty" and "sampling uncertainty" the authors did not use the keyword till at the very end of the description/definition of it. E.g. "Finally, specifying a model and parameter values is not sufficient to run the actual computations – a specific implementation and computational method must be chosen, or even developed, before a statistical model can be estimated or predictions from a mechanistic or agnostic model derived. Again, there is a multitude of options without clear guidance or a definitive choice on the method that will provide the most suitable answer to their research question: researchers encounter here *method uncertainty* [55]." While this might seem elegant, it is very difficult for the reader to follow. I would suggest always to use the keyword as early as possible and then subsequently define it in the following sentence(s).
3. I found Figure 3 difficult to understand with respect to its description in the text. The text states the following: "Taking confidence intervals as an example, Figure 3 illustrates how they can be just as prone as p-values to selective reporting.". However, Figure 3 and its caption does not mention confidence intervals. I felt like this was the weakest figure in the manuscript.
4. There were times in the paper when I wanted a table to summarise lists presented in paragraphs. For example, for the six types of uncertainty. While Figure 2 did a good job in illustrating where in the research process the types of uncertainty occur, I think a table with short descriptions for each type of uncertainty would be very helpful for the reader to refer to. Furthermore, I am unsure whether Figures 4 and 5 need to be figures; I think tables might be more user friendly (especially for Figure 4), especially as you can add short descriptions in a table as well.
5. On page 10, line 49 the authors talk about cumulative knowledge and multiple lines of evidence. I was wondering whether they would want to mention the keyword "triangulation" as that is frequently used by some areas of science to describe this process: Munafò, Marcus R., and George Davey Smith. 'Robust Research Needs Many Lines of Evidence'. *Nature* 553, no. 7689 (January 2018): 399–401. <https://doi.org/10.1038/d41586-018-01023-3>.
6. Page 10, line 56: I think "Steps to make ones Replicable" would be better as its own section, rather than part of the list ("e") as it is not in Figure 4.

With best wishes,
Dr Amy Orben

Decision letter (RSOS-201925.R0)

Dear Dr Hoffmann

On behalf of the Editors, we are pleased to inform you that your Manuscript RSOS-201925 "The multiplicity of analysis strategies jeopardizes replicability: lessons learned across disciplines" has been accepted for publication in Royal Society Open Science subject to minor revision in accordance with the referees' reports. Please find the referees' comments along with any feedback from the Editors below my signature.

Please submit your revised manuscript and required files (see below) no later than 7 days from today's (ie 02-Mar-2021) date. Note: the ScholarOne system will 'lock' if submission of the revision is attempted 7 or more days after the deadline. If you do not think you will be able to meet this deadline please contact the editorial office immediately.

on behalf of Professor Zoltan Dienes (Associate Editor) and Mark Chaplain (Subject Editor)
openscience@royalsociety.org

Reviewer comments to Author:
Reviewer: 1

Comments to the Author(s)

The manuscript integrates understanding across multiple disciplines to present a framework around analysis multiplicity. It does so with, what I think is, complete success. I thought this manuscript was carefully written, clearly argued and of importance to a broad scientific audience. I would therefore recommend it for publication.

There are a couple of minor points that the authors can decide whether they think would improve the manuscript:

1. I would suggest the authors might want to lengthen their figure headings slightly (e.g. Fig. 1 and 2), as readers often read those separately to the paper and at the moment they are not self-explanatory.
2. On page 7 there were two instances where the structure of sentences made the argument less clear than it could have been. For both "method uncertainty" and "sampling uncertainty" the authors did not use the keyword till at the very end of the description/definition of it. E.g. "Finally, specifying a model and parameter values is not sufficient to run the actual computations – a specific implementation and computational method must be chosen, or even developed, before a statistical model can be estimated or predictions from a mechanistic or agnostic model derived. Again, there is a multitude of options without clear guidance or a definitive choice on the method that will provide the most suitable answer to their research question: researchers encounter here *method uncertainty* [55]." While this might seem elegant, it is very difficult for the reader to follow. I would suggest always to use the keyword as early as possible and then subsequently define it in the following sentence(s).
3. I found Figure 3 difficult to understand with respect to its description in the text. The text states the following: "Taking confidence intervals as an example, Figure 3 illustrates how they can be just as prone as p-values to selective reporting.". However, Figure 3 and its caption does not mention confidence intervals. I felt like this was the weakest figure in the manuscript.
4. There were times in the paper when I wanted a table to summarise lists presented in paragraphs. For example, for the six types of uncertainty. While Figure 2 did a good job in illustrating where in the research process the types of uncertainty occur, I think a table with short descriptions for each type of uncertainty would be very helpful for the reader to refer to. Furthermore, I am unsure whether Figures 4 and 5 need to be figures; I think tables might be more user friendly (especially for Figure 4), especially as you can add short descriptions in a table as well.
5. On page 10, line 49 the authors talk about cumulative knowledge and multiple lines of evidence. I was wondering whether they would want to mention the keyword "triangulation" as that is frequently used by some areas of science to describe this process: Munafò, Marcus R., and George Davey Smith. 'Robust Research Needs Many Lines of Evidence'. *Nature* 553, no. 7689 (January 2018): 399–401. <https://doi.org/10.1038/d41586-018-01023-3>.
6. Page 10, line 56: I think "Steps to make ones Replicable" would be better as its own section, rather than part of the list ("e") as it is not in Figure 4.

With best wishes,
Dr Amy Orben

===PREPARING YOUR MANUSCRIPT===

Please ensure that you include an acknowledgements' section before your reference list/bibliography. This should acknowledge anyone who assisted with your work, but does not

qualify as an author per the guidelines at <https://royalsociety.org/journals/ethics-policies/openness/>.

===PREPARING YOUR REVISION IN SCHOLARONE===

-- Ensure that your data access statement meets the requirements at <https://royalsociety.org/journals/authors/author-guidelines/#data>. You should ensure that you cite the dataset in your reference list. If you have deposited data etc in the Dryad repository, please only include the 'For publication' link at this stage. You should remove the 'For review' link.

-- If you have uploaded ESM files, please ensure you follow the guidance at <https://royalsociety.org/journals/authors/author-guidelines/#supplementary-material> to include a suitable title and informative caption. An example of appropriate titling and captioning may be found at https://figshare.com/articles/Table_S2_from_Is_there_a_trade-off_between_peak_performance_and_performance_breadth_across_temperatures_for_aerobic_sc_ope_in_teleost_fishes_/3843624.

Author's Response to Decision Letter for (RSOS-201925.R0)

See Appendix A.

Decision letter (RSOS-201925.R1)

Dear Dr Hoffmann,

It is a pleasure to accept your manuscript entitled "The multiplicity of analysis strategies jeopardizes replicability: lessons learned across disciplines" in its current form for publication in Royal Society Open Science.

Please see the Royal Society Publishing guidance on how you may share your accepted author manuscript at <https://royalsociety.org/journals/ethics-policies/media-embargo/>. After

publication, some additional ways to effectively promote your article can also be found here <https://royalsociety.org/blog/2020/07/promoting-your-latest-paper-and-tracking-your-results/>.

on behalf of Professor Zoltan Dienes (Associate Editor) and Mark Chaplain (Subject Editor)
openscience@royalsociety.org

Appendix A

Response to Reviewers: The multiplicity of analysis strategies jeopardizes replicability: lessons learned across disciplines

March 9, 2021

Reviewer 1 Dr Amy Orben

The manuscript integrates understanding across multiple disciplines to present a framework around analysis multiplicity. It does so with, what I think is, complete success. I thought this manuscript was carefully written, clearly argued and of importance to a broad scientific audience. I would therefore recommend it for publication.

We are grateful to the referee for her careful evaluation of our submission, her valuable comments and her helpful suggestions, which contributed to a significant improvement of the quality of the paper. All comments have been addressed, and the manuscript has been revised accordingly.

There are a couple of minor points that the authors can decide whether they think would improve the manuscript:

1. I would suggest the authors might want to lengthen their figure headings slightly (e.g. Fig. 1 and 2), as readers often read those separately to the paper and at the moment they are not self-explanatory.
✓ Thank you for this suggestion. We revised the manuscript accordingly.
2. On page 7 there were two instances where the structure of sentences made the argument less clear than it could have been. For both method uncertainty and sampling uncertainty the authors did not use the keyword till at the very end of the description/definition of it. E.g. Finally, specifying a model and parameter values is not sufficient to run the actual computations a specific implementation and computational method must be chosen, or even developed, before a statistical model can be estimated or predictions from a mechanistic or agnostic model derived. Again, there is a multitude of options without clear guidance or a definitive choice on the method that will provide the most suitable answer to their research question: researchers

encounter here *method uncertainty* [55]. While this might seem elegant, it is very difficult for the reader to follow. I would suggest always to use the keyword as early as possible and then subsequently define it in the following sentence(s).

✓ *Thank you for these helpful suggestions. We revised the manuscript accordingly.*

3. I found Figure 3 difficult to understand with respect to its description in the text. The text states the following: "Taking confidence intervals as an example, Figure 3 illustrates how they can be just as prone as p-values to selective reporting.". However, Figure 3 and its caption does not mention confidence intervals. I felt like this was the weakest figure in the manuscript.

✓ *Thank you for this comment. We revised the caption of Figure 3 accordingly.*

4. There were times in the paper when I wanted a table to summarise lists presented in paragraphs. For example, for the six types of uncertainty. While Figure 2 did a good job in illustrating where in the research process the types of uncertainty occur, I think a table with short descriptions for each type of uncertainty would be very helpful for the reader to refer to. Furthermore, I am unsure whether Figures 4 and 5 need to be figures; I think tables might be more user friendly (especially for Figure 4), especially as you can add short descriptions in a table as well.

✓ *Thank you for these suggestions. We added Table 1 to provide short descriptions for each type of uncertainty and replaced Figure 5 with Table 2. Concerning Figure 4, we think that the implementation that is most helpful for readers is to keep it as a Figure but to provide a link to an interactive and clickable webpage where they can find additional references for each solution.*

5. On page 10, line 49 the authors talk about cumulative knowledge and multiple lines of evidence. I was wondering whether they would want to mention the keyword "triangulation" as that is frequently used by some areas of science to describe this process: Munaf, Marcus R., and George Davey Smith. Robust Research Needs Many Lines of Evidence. *Nature* 553, no. 7689 (January 2018): 399401. <https://doi.org/10.1038/d41586-018-01023-3>.

✓ *Thank you for this helpful comment. We have revised the manuscript accordingly.*

6. Page 10, line 56: I think "Steps to make ones Replicable" would be better as its own section, rather than part of the list ("e") as it is not in Figure 4.

✓ *Thank you for pointing this out. We have revised the manuscript accordingly.*